# The N-Linked Glycosylation Site N191 Is Necessary for PKA Signal Transduction in Eel Follicle-Stimulating Hormone Receptor

**DOI:** 10.3390/ijms232112792

**Published:** 2022-10-24

**Authors:** Munkhzaya Byambaragchaa, Hong-Kyu Park, Dae-Jung Kim, Jong-Hyuk Lee, Myung-Hwa Kang, Kwan-Sik Min

**Affiliations:** 1Institute of Genetic Engineering, Hankyong National University, Ansung 17579, Korea; 2Animal Biotechnology, Graduate School of Future Convergence Technology, Hankyong National University, Ansung 17579, Korea; 3Aquaculture Industry Division, South Sea Fisheries Research Institute, National Institute of Fisheries Science (NIFS), Yeosu 59780, Korea; 4College of Pharmacy, Chung-Ang University, Seoul 06974, Korea; 5Department of Food Science and Nutrition, Hoseo University, Asan 31499, Korea; 6Carbon-Neutral Resources Research Center, Hankyong National University, Ansung 17579, Korea

**Keywords:** eel FSHR, GPCR signaling, N-linked glycosylation site, cAMP pathway, loss of cell surface receptor

## Abstract

The follicle-stimulating hormone receptor (FSHR) contains several N-linked glycosylation sites in its extracellular region. We conducted the present study to determine whether conserved glycosylated sites in eel FSHR are necessary for cyclic adenosine monophosphate (cAMP) signal transduction. We used site-directed mutagenesis to induce four mutations (N120Q, N191Q, N272Q, and N288Q) in the N-linked glycosylation sites of eel FSHR. In the eel FSHR wild-type (wt), the cAMP response was gradually increased in a dose-dependent manner (0.01–1500 ng/mL), displaying a high response (approximately 57.5 nM/10^4^ cells) at the Rmax level. Three mutants (N120Q, N272Q, and N288Q) showed a considerably decreased signal transduction as a result of high-ligand treatment, whereas one mutant (N191Q) exhibited a completely impaired signal transduction. The expression level of the N191Q mutant was only 9.2% relative to that of the eel FSHR-wt, indicating a negligible expression level. The expression levels of the N120Q and N272Q mutants were approximately 35.9% and 24% of the FSHG-wt, respectively. The N288Q mutant had an expression level similar to that of the eel FSHR-wt, despite the mostly impaired cAMP responsiveness. The loss of the cell surface agonist-receptor complexes was very rapid in the cells expressing eel FSHR-wt and FSHR-N288Q mutants. Specifically, the N191Q mutant was completely impaired by the loss of cell surface receptors, despite treatment with a high concentration of the agonist. Therefore, we suggest that the N191 site is necessary for cAMP signal transduction. This finding implies that the cAMP response, mediated by G proteins, is directly related to the loss of cell surface receptors as a result of high-agonist treatment.

## 1. Introduction

The follicle-stimulating hormone (FSH) receptor belongs to the subfamily of G protein-coupled receptors (GPCRs) that are phosphorylated in the intracellular and carboxyl-terminal regions [1]. The responsiveness of receptor cells to hormones decreases over time, despite continuous ligand exposure. This phenomenon, known as desensitization, is caused by factors that regulate the quantity of the hormone receptor [2]. The receptors occupied by the agonist respond to the G protein stimulator (Gs), which, in turn, activates adenylyl cyclase, resulting an in increased synthesis of cyclic adenosine monophosphate (cAMP) through the protein kinase A pathway (PKA) [3,4,5].

All GPCRs have a transmembrane domain that consists of seven-membrane-traversing α-helices connected by three extracellular and three intracellular loops [6]. The intracellular cAMP levels are increased in cells expressing glycoprotein hormone receptors following constitutive ligand treatment [3,7]. Therefore, cAMP controls the response of the cAMP-dependent PKA pathway, which is necessary for various downstream cellular processes [8,9]. Thus, GPCRs are important molecules that mediate various physiological processes [10].

In fish, FSH and luteinizing hormones (LH) are known to regulate the early phases of gametogenesis, oocyte maturation, and ovulation [11,12]. The fish gonadotropin hormone (GTH) receptors (FSHR and LHR) have been characterized in African catfish [13,14] and zebrafish [15,16], demonstrating that both of these FSHR- and LHR-expressing cells respond to their ligands in vitro.

We have studied the GPCR signal transduction of activating and inactivating lutropin/chorionic gonadotropin receptors (LH/CG) and FSHRs, showing that the highly conserved regions in glycoprotein hormone receptors are similarly regulated in different species of both mammals and fish [3,4,5]. In recent years, we have been studying signal transduction in activating and inactivating mutants to characterize the specific function of the PKA pathway [17] and cell-surface loss of receptors [18] in eel LH/CGR. We found that the phosphorylation sites in the C-terminal region of equine LH/CGR are important for signal transduction [19]. The ligand glycosylation sites of eel FSH [8,9], eel LH [20], and equine CG [21,22,23,24,25] demonstrate that the N-linked glycosylation sites are important for signal transduction in cells expressing these receptors. These results have elucidated the effects of diverse agonists on signal transduction in eel FSHR, eel LH/CGR, and equine LH/CGR, but little is known about the effect of N-linked glycosylation on signal transduction in the glycoprotein hormone receptors of fish species, including FSHR and LH/CGR.

The present study aimed to determine the functions of the glycosylation sites in highly conserved regions of glycoprotein hormone receptors, including FSHR and LH/CGR. We analyzed the response of the PKA signaling pathway to cAMP and the loss of cell-surface receptors. Our results revealed that cAMP signal transduction, expression, and loss of cell surface receptors in the N191 amino acid play a very important role in cells expressing these mutants.

## 2. Results

### 2.1. Preparation and Cell-Surface Expression of Eel FSHR Mutants and Expression Vectors

To determine how N-linked glycosylation sites of eel FSHR affect the hormone-receptor interactions, we generated four mutations (N120Q, N191Q, N272Q, and N288Q) in the extracellular domain of eel FSHR, as shown in Figure 1. The surface expression of eel FSHR mutants was determined in transiently transfected human embryonic kidney (HEK) 293 cells, using an enzyme-linked immunosorbent assay (ELISA) (Figure 2). The sites N191 and N272 are well-conserved regions between mammals and fish, whereas the N120 and N288 N-linked glycosylation sites are specific compared to mammalian FSHRs. A deletion of approximately 30 aa exists between the N288 glycosylation site and transmembrane domain 1 in eel FSHR, as described previously in our lab [3]. The expression level of eel FSHR-wt was considered to be 100%, and the expression level of the N288Q mutant was very similar to that of FSHR-wt, whereas the expression levels of the other three mutants were significantly decreased to 35% or less of the FSHR-wt. The expression levels of the N120Q and N272Q mutants were 35.9% and 24.0%, respectively, whereas the expression of the N191Q mutant was only 9.0%. The specific N-linked glycosylation site N191Q in the eel FSHR had a remarkable effect on the cell surface expression in the cells expressing the mutant. Subsequently, we determined the cAMP response and cell surface receptor loss induced by agonist treatment.

### 2.2. cAMP Responsiveness Induced by Agonist in N-Linked Glycosylated Mutants

The effects of the glycosylated mutants on basal and ligand-stimulated cAMP responsiveness are summarized in Figure 3 and Table 1. The transfected cells were analyzed for the cAMP response following an agonist-mediated induction. There was no difference in the basal cAMP response between the eel FSHR-wt and all of the mutants. The cAMP production increased in a dose-dependent manner (Figure 3). The Rmax cAMP level in the wild-type receptor was 57.5 nM/10^4^ cells, and the half-maximal effective concentration (EC_50_) of ligand-stimulated cAMP was approximately 201.1 ng/mL.

In the N120Q mutant, the plot moved to the right compared to that of eel FSHR-wt, indicating very low EC_50_ values of approximately 283.3 ng/mL. The EC_50_ value for the cAMP production was approximately 0.7-fold higher than that of the eel FSHR-wt. The Rmax value was 25.9 ± 1.9 nM, displaying that the Rmax values were approximately 45% of those of the eel FSHR-wt. The EC_50_ values in the cells expressing the N272Q and N288Q mutants were 326.8 ng/mL and 281 ng/mL, respectively. These values indicated 0.61- and 0.71-fold decreases, respectively. The Rmax levels in those two mutants were 19.2 nM/10^4^ cells and 21.4 nM/10^4^ cells, representing from 33.4% to 37.2% of that of eel FSHR-wt, respectively (Figure 4). Thus, these three mutants (N120Q, N272Q, and N288Q) had considerably lower EC_50_ and Rmax values compared to the FSHR-wt. We suggested that two of the mutants (N120Q and N272Q) may be due to the low expression level on the cell surface, but the N288Q mutant was fully expressed on the cell surface. We suggest that these glycosylation sites are involved in signal transduction, as indicated by the markedly decreased EC_50_ and Rmax levels of the mutants. Specifically, the cAMP response of the N191Q mutant was completely inhibited, demonstrating that the signal transduction for cAMP responsiveness was not activated by the high-concentration ligand treatment in the cells expressing the N191Q mutant. The expression levels of the N191Q mutant were dramatically decreased, to 9%, compared to that of eel FSHR-wt. Thus, we also suggest that the completely impaired cAMP response is involved in the expression level on the cell surface. Therefore, the N191 site in the N-linked glycosylation site of the eel FSHR plays a pivotal role in the cAMP response.

### 2.3. Cell-Surface Receptor Loss Induced by Treatment with the Eel FSH Agonist

Next, we used ELISA to accurately analyze the rate of cell surface receptor loss in order to elucidate the relationship between cAMP responsiveness and cell-surface loss. The cell surface receptor loss was measured in a time-dependent manner, and the results are shown in Figure 5.

The cell surface receptor expression in the eel FSHR-wt decreased until it reached approximately 68% at 15 min post-treatment. The level was maintained at approximately 70% for 240 min. The cells expressing the eel FSHR-N272Q mutants displayed a very similar expression level of 69% of the FSHR-wt at 15 min, and the level subsequently decreased to 64% at 30 min. Subsequently, this value increased to 80% after 60 min. In the other two mutants, FSHR-N120Q and N288Q, the cell surface receptor expression decreased to 78% and 73% of FSHR-wt in the first 15 min, respectively. After that, the expression level of the N120Q mutant slowly increased and subsequently exhibited 94% at 240 min, demonstrating that the cell surface receptor loss no longer occurred, despite the high- concentration agonist treatment. However, the cells expressing the N288Q mutant displayed a pattern similar to that of the eel FSHR-wt until the final detection time.

Specifically, the surface loss of the receptor was not observed in the cells expressing the eel FSHR-N191Q mutant. The loss of cell surface receptors at 30 min decreased remarkably in the eel FSHR-wt (30%) compared to that in the cells without the agonist treatment (0% cell surface receptor loss) (Figure 6). In the cells harboring the N120Q and N288Q mutants, the surface loss of the receptors was decreased by 23% and 26%, respectively, compared to that observed in the eel FSHR-wt at 30 min post-agonist treatment.

However, the cells expressing the N272Q mutant displayed an increase of 36% in the surface receptor loss compared to the eel FSHR-wt. These data clearly show that the cell surface receptor loss in the two mutants (eel FSHR-N120Q and N272Q) was similar to that of the eel FSHR-wt at 30 min. However, the expression levels of the N120Q and N272Q mutants were only 24–35% of that of the eel FSHR-wt. The N191Q mutant displayed almost no receptor expression at the cell surface, suggesting that the cAMP responsiveness and cell surface loss were not induced by the high-concentration agonist treatment. The Eel FSHR-wt and N288Q mutants sustained their rates of cell surface loss until the final detection time, whereas the N288Q mutant did not show any cAMP signal responsiveness.

## 3. Discussion

The present study describes the signal transduction functions of the N-linked glycosylation sites located in the extracellular domain of eel FSHR. Our findings demonstrate that the N191Q mutation at the specific glycosylation site completely impaired the cAMP signal transduction, suggesting that the N191Q mutant is rarely expressed on the surface of HEK-293 cells and does not cause cell surface loss of the receptor. Our recent observations have suggested that activating or inactivating conformations of eel FSHR [3], equine FSHR [5], and equine LH/CGR [4] are significantly involved in the signal transduction of G proteins and the cell-surface loss of receptors by high-concentration agonist treatment. The N-linked glycosylation sites in the glycoprotein hormones are important for PKA signal transduction in the cells expressing these receptors [17,18,19,20,21,22,23,24,25].

In the present study, we characterized the signal transduction of four N-linked glycosylation sites in the extracellular domain region of eel FSHR, which are highly conserved in FSHR, including mammalian and fish species. Our results showed that the N191Q mutation in the eel FSHR dramatically attenuated the agonist-induced cAMP responsiveness through the receptors, suggesting that the glycosylation site causes impaired activation. Compared to the eel FSHR-wt, the N120Q, N272Q, and N288Q mutations resulted in a 0.61- to 0.71-fold decrease in the EC_50_ value, demonstrating that these mutations affect the cAMP signal transduction pathway, as previously reported for mammalian FSHRs and LH/CGRs [26,27,28,29,30]. These results are consistent with those of a previous study, which found that the N174Q mutation in rat FSHR was required for the efficient folding of the nascent receptor to allow the high-affinity binding of the FSH hormone ligand [28]. In rat LH/CGR, the same glycosylation site in the conserved domain region is necessary, indicating that the site is involved in the high-affinity binding of hormones [27]. Another study reported that N-linked oligosaccharides are not absolutely required for the proper folding of rat LH/CGR for ligand binding and signal transduction, but it also suggested that rat FSHR strictly requires N-linked glycosylation sites for folding [29]. In the human thyrotropin receptor, N-linked oligosaccharide sites at Asn77 and Asn113 play a role in the expression of biologically active receptors on the cell surface [31]. Our results also suggest that all sites of the N-linked glycosylation are not necessary for signal transduction in cAMP responsiveness. 

We identified two mutants (N120Q and N272Q) for N-glycosylation, which suggests that modification is important for the surface expression and functional response, indicating that those mutations were modified as inactive conformations. However, the N228Q mutant resulted in a dramatically decreased cAMP response despite fully expressing on the cell surface. Therefore, we suggest that the stability and conformation of the receptors are recognized as extremely important in the cAMP signal transduction. The specific site of Asn191 appears to be related to the signal transduction pathway. We also assume that the mutation at this site may result in a lower expression on the cell surface, suggesting that the cAMP response and cell surface loss of the receptor are completely impaired.

In recent studies, we reported that the mutations D540G and D540N in the eel FSHR are constitutively activating, whereas the inactivating mutations A193V, N195I, R546C, and A548V completely impaired the cAMP response [3]. These mutants showed similar expression levels, but the expression of the cell surface receptors decreased to approximately 70–80% in the first 5 min after the agonist treatment. We also reported that equine LH/CGR and equine FSHR with activating mutations exhibited highly increased basal cAMP responses, and consequently, the cell surface loss of the receptors showed a higher increase of approximately 55% [4,5]. These results are consistent with our current data, demonstrating that specific conserved amino acids at the glycosylation sites are necessary for signal transduction in the PKA pathway. Therefore, we suggest that these differences in the PKA pathway in the glycoprotein hormone receptors probably altered post-translation receptor folding.

Other studies have suggested that the N-linked glycosylation sites Asn291 and Asn369 of the thyrotropin receptor are related to the cleavage or non-cleavage [32], the hormone-binding activity of human LH/CGR [33], and the proper folding of the human LH/CG receptor [34]. Mutations in the other GPCR glycosylation site, GPR 176 (also known as HB-954; a class-A orphan GPCR), containing four conserved N-linked glycosylation residues in the N-terminal region, reduced the protein expression and attenuated the cAMP responsiveness in the cells [35]. The N-glycosylation of the CPCRs is very important for the folding, ER exit [36], and cell surface expression of the SARS-CoV-2 receptor ACE2 [37]. The site Asn72 in the human secretin receptor consists of five N-linked glycosylation sites, which alternatively reduce the maximal responses, suggesting that this site is a true glycosylation signal [38]. The GPRC6A receptor, leading to a Gq-coupled response and production of inositol phosphate messenger molecules, modulates the surface expression at the Asn86 and Asn555 sites, and some of these residues regulate the functional signaling of the receptor [39].

Therefore, we demonstrated that the N-linked glycosylation sites in the GPCR are associated with site-specific functions (cell surface expression and PKA/PKC signal transduction), demonstrating that mutations in the specific site, Asn191, in the eel FSHR completely impair the cAMP responsiveness, cell surface expression, and cell surface loss of receptors by the high-concentration agonist treatment. Thus, our results demonstrate that N-linked glycosylation sites in the eel FSHR are required for the effective expression on the cell surface and PKA signal transduction.

In human FSHR, mutations L501F and I505V resulted in low cAMP production and ERK phosphorylation, demonstrating that these residues in the extracellular loop 2 regions are important in the FSH-mediated signaling pathways [40]. The researchers also reported that K589N and A590S substitutions impaired pERK1/2 activity [41]. An FSH stimulation in granulosa cells leads to pERK1/2 activation by PKA, which in turn phosphorylates the downstream signals of the MAPK signal transduction pathway [42]. Other studies have reported that the downstream signal transduction by FSH stimulation activates the MAPK/ERK pathway [43], whereas the G protein-coupled receptor kinases (GRK) 5 and GRK6 are required for ERK activation [44]. Although the pERK1/2 data were not shown, we attempted to determine the relationship between the PKA signal transduction and pERK1/2 activity, showing that the cAMP signal transduction was completely downregulated in the cells expressing the eel FSHR-N191Q mutant. Therefore, we suggest that the pERK1/2 signal transduction in glycoprotein hormone receptors, including FSHR, requires further elucidation.

Therefore, based on our results, the N-linked glycosylation sites investigated herein are not categorically required for PKA signal transduction, whereas the specific site, Asn191, is necessary for the functioning of the AMP signaling pathway. Although the present study was restricted to cells expressing eel FSHR-mutants, our data show that PKA signal transduction depends on the cell surface expression of the GPCR receptors. In our future studies, we will try to elucidate the different signaling pathways involved in the β-arrestin recruitment, GRK activation, and pERK1/2 in HEK 293 cells.

## 4. Materials and Methods

### 4.1. Materials

The pGEM-T cloning vector was purchased from Promega (Madison, WI, USA). The oligonucleotides for the PCR mutations were synthesized by Genotech (Daejeon, Korea). The DNA ligation kit, restriction enzymes, and polymerase chain reaction (PCR) reagents were obtained from Takara (Shiga, Japan) and Toyobo (Osaka, Japan). The pcDNA3 vector, FreeStyle MAX reagent, FreeStyle CHO-suspension (CHO-S) cells, Lipofectamine-2000, and Lipofectamine-3000 were purchased from Invitrogen (Carlsbad, CA, USA). The pCORON1000 SP VSV-G tag expression vector was provided by GE Healthcare (Chicago, IL). The Opti-MEM, OptiPRO, FreeStyle CHO expression medium, serum-free CHO-S-SFM II, and Ham’s F-12 medium were purchased from Gibco BRL (Grand Island, NY, USA). The HEK 293 cells and CHO-K1 cells were obtained from the Korean Cell Line Bank (KCLB, Seoul, Korea). The DNA purification kits, QIAprep Spin plasmid kits, and QIAGEN Maxi plasmid kits were purchased from Qiagen, Inc. (Hilden, Germany). The centrifugal filter devices (Amicon Ultra-4 10 K) for the concentration of the recombinant eel FSH hormone were purchased from Amicon Bio (Billerica, MA, USA). The monoclonal antibodies (5A11, 11A8, and 14F5) for the recombinant eel FSH analysis were produced in our laboratory, as previously reported [8]. The SuperSignal enzyme-linked immunosorbent assay (ELISA) Femto Maximum substrate was purchased from Thermo Fisher Scientific (Waltham, MA, USA). The horseradish peroxidase (HRP) for the labeling of the primary antibodies was generously donated by Medexx Inc. (Seongnam, Kyeong-gi, Korea). The tethered eel FSH β/α cDNA and eel FSHR cDNAs were cloned from pituitary and ovary/testis cDNA, as previously reported by us [3,8,9]. The cAMP Dynamic 2 HTRF Assay Kit was purchased from Cisbio (Codolet, France). The glass spinner and disposable flasks were provided by Corning Inc. (Corning, NY, USA). All of the other reagents were purchased from Sigma-Aldrich (St. Louis, MO, USA) and Wako Pure Chemical Industries (Osaka, Japan).

### 4.2. Site-Directed Mutagenesis

The schematic representation of the four glycosylation sites in the extracellular domain region of the eel FSHR-wt (120, 191, 272, and 288) is shown in Figure 1. First, the point mutation primers for the glycosylation sites were designed and synthesized. Each glycosylation site was changed from Asn (AAC) to Gln (CAG) by using an overlapping PCR strategy. First, two fragments were generated, one by performing PCR with a forward primer, including the XhoI site and a mutant reverse primer, and the other using a forward mutant primer and a reverse primer including the EcoRI site. Subsequently, the second PCR was performed to produce full-length mutants with two templates using the forward and reverse primers. After the PCR products were produced, the full-length fragments were ligated into the pGEM-T easy vector, and the sequence of the entire region of each mutant generated by the PCR was confirmed by DNA sequencing. Finally, we constructed five receptor genes: eel FSHR-wt, eel FSHR-N120Q, eel FSHR-N191Q, eel FSHR-N272Q, and eel FSHR-N288Q.

### 4.3. Vector Construction for Transfection in Mammalian Cells

Fragments were generated using the XhoI and EcoRl restriction enzymes. The fragments were subcloned into the expression vector pCORON1000 SP VSV-G or the pcDNA3 vector by cutting the XhoI and EcoRl sites. The vector and target DNA were added at a ratio of 1:3, mixed with 2× concentrated buffer, and incubated overnight at 16 °C. The inserts were then confirmed by cutting with the XhoI and EcoRl enzymes. Finally, 10 expression vectors were constructed in the pVSVG and pcDNA3 (designated as pVSVG-eel FSHR-wt, pVSVG-eel FSHR-N120Q, pVSVG-eel FSHR-N191Q, pVSVG-eel FSHR-N272Q, and pVSVG-eel FSHR-N288Q; pcDNA3-eel FSHR-wt, pcDNA3-eel FSHR-N120Q, pcDNA3-eel FSHR-N191Q, pcDNA3-eel FSHR-N272Q, and pcDNA3-eel FSHR-N288Q.

### 4.4. Production and ELISA Analysis of Recombinant-eel FSH (rec-eel FSH)

For the recombinant eel FSH hormone production, the plasmids cloned into the pcDNA3 expressing vector were transfected into CHO-S cells using the FreeStyle™ MAX reagent method, as previously described [3]. The CHO-S cells were passaged at a density of 5 × 10^5^ cells/mL for one day before transfection. The cells were cultured in a spinner flask on an orbital shaker platform at 120–130 rpm and 37 °C, in an environment containing 8% CO_2_. The next day, the cell density was approximately 1 × 10^6^ cells/mL. The plasmid DNA (260 µg) and 260 µL of FreeStyle MAX reagent were combined. The total volume was diluted using OptiPro SFM. After 10 min of incubation at room temperature, the complex samples were slowly added to the cell culture medium. The transfected cells were cultured for seven days, and the supernatants were collected by centrifugation at 100,000× *g* for 10 min at 4 °C to remove the cell debris. The samples were concentrated by approximately 20- fold, using a centrifugal filter. The concentrated samples were mixed with phosphate-buffered saline (PBS), and the quantity of the recombinant eel FSH was analyzed using a double-sandwich ELISA developed previously in our laboratory [8].

### 4.5. Transient Transfection of Eel FSHR Mutant Genes

The transient transfection of the CHO-K1 and HEK 293 cells was conducted using the liposome method, as previously reported [3]. The CHO-K1 cells were cultured in the normal growth medium [Ham’s F-12 medium containing penicillin (50 U/mL), streptomycin (50 µg/mL), glutamine (2 mM), and 10% fetal bovine serum]. The HEK 293 cells were cultured in the normal growth medium [DMEM containing HEPES (10 mM), gentamycin (50 µg/mL), and 10% FBS]. One day before the transfection, the cells were seeded in a six-well plate at 2 × 10^5^ cells/well (CHO-K1) and 5 × 10^5^ cells/well (HEK 293). The next day, most of the cells had grown to 80–90% confluency and were transfected with the eel FSHR mutant plasmids (2.5 ug) using lipofectamine (5 µL). The transfected wells were supplemented with a growth medium containing 20% FBS at 5 h post-transfection. Subsequently, the cells were seeded in 96-well and 384-well plates. Finally, the cAMP levels were analyzed 48–72 h post-transfection.

### 4.6. cAMP Analysis by Homogeneous Time-Resolved Fluorescence (HTRF)

The cAMP accumulation in the CHO-K1 cells expressing the eel FSHR-wt and mutants was measured using cAMP Dynamic 2 competitive immunoassay kits, as described previously [3,4]. Briefly, the assay was conducted using two cAMP antibodies labeled with cryptate-conjugated anti-cAMP and d2. The transfected CHO-K1 cells were diluted in 0.5 mM IBMX (cAMP degradation inhibitor) and seeded in 384-well plates (10,000 cells per well). Standard samples were prepared to cover an average cAMP concentration (0.17–712 nM). Each well was supplemented with 5 µL of recombinant eel FSH and the plate was sealed. The plate was incubated for cell stimulation at RT for 30 min. The detection reagents cAMP-d2 and anti-cAMP-cryptate were added, followed by incubation at RT for 1 h. The cAMP production was detected by measuring the decrease in the homogeneous time-resolved fluorescence (HTRF) energy transfer (665 nm/620 nm) using a Tristar2 S LB942 microplate reader (BERTHOLD Tech, Wildbad, Germany). The specific signal Delta F % (energy transfer) was inversely proportional to the concentration of the cAMP in the standard or sample. The results were calculated from the 665 nm/620 nm ratio and expressed as ΔF% (cAMP inhibition) according to the following equation: [ΔF% = (standard or sample ratio − negative ratio) × 100/ sample negative ratio]. The cAMP concentrations corresponding to the ΔF% values were calculated using GraphPad Prism software (version 6.0; GraphPad Software, Inc., La Jolla, CA, USA).

### 4.7. Agonist-Induced Loss of Cell Surface Receptor

The cell surface loss of the eel FSHR was assessed using ELISA, as described previously [3]. The cells were plated at a density of 6 × 10^5^ cells per 60 mm dish and then transfected with the eel FSHR-wt and mutants. The transfected cells were split into 96-well dishes (1 × 10^4^ cells), coated with poly-D-lysine, 24 h post-transfection. The cells were pre-incubated with the recombinant eel FSH (1000 ng/mL) for the time-dependent tests (5, 15, 60, 120, and 240 min). The cells were fixed with 4% paraformaldehyde in Dulbecco’s PBS (DPBS) for 5 min. After washing twice with the DPBS, the wells were incubated with a blocking solution (Tris-buffered saline with 1% bovine serum albumin) for 30 min. The primary antibody was reacted with rabbit anti-VSVG antibody (1:1000), followed by incubation with horseradish peroxidase-conjugated anti-rabbit antibody (1:500). After washing four times with the blocking solution, 80 μL of the PBS and 10 μL of the SuperSignal ELISA Femto Maximum substrate were added to each well for detection. The luminescence was measured using a Cytation 3 plate reader (BioTek, Winooski, VT, USA). The expression level of the eel FSHR-wt was considered 100%. The loss of the cell surface receptors was calculated by comparing the levels in the presence of the recombinant eel FSH to the levels in the untreated cells (taken as 0% of the loss of the cell surface receptors).

### 4.8. Data Analysis

A sequence alignment was performed using the MultAlin multiple sequence analysis tool. GraphPad Prism 6.0 (San Diego, CA, USA) was used to analyze the cAMP and EC_50_ values. The curves fitted in a single experiment were normalized to the background signal measured in mock-transfected cells (Figure 3). A one-way analysis of variance (ANOVA) and Tukey’s multiple comparison tests were used to compare the results between the samples using GraphPad Prism 6.0 software. A *p*-value of <0.05 was considered to indicate a significant difference between the groups.

## 5. Conclusions

This study demonstrated that N120Q, N191Q, N272Q, and N288Q mutations in the N-linked glycosylation sites of eel FSHR resulted in a decrease in cAMP production, and the mutation at the specific site, N191, resulted in a completely impaired signal transduction of the agonist-mediated receptor response, caused by an expression level of only 9% of that of the eel FSHR-wt. The results clearly show that the specific N191Q mutation at the N-linked glycosylation site significantly affected the cell surface expression. The cell surface loss of the receptor in the N191Q mutant was not reversed by the high-agonist treatment for 240 min. These findings improve our understanding of FSHR function and regulation with respect to mutations in the N-linked glycosylation sites in mammalian glycoprotein hormone receptors.

## Figures and Tables

**Figure 1 ijms-23-12792-f001:**
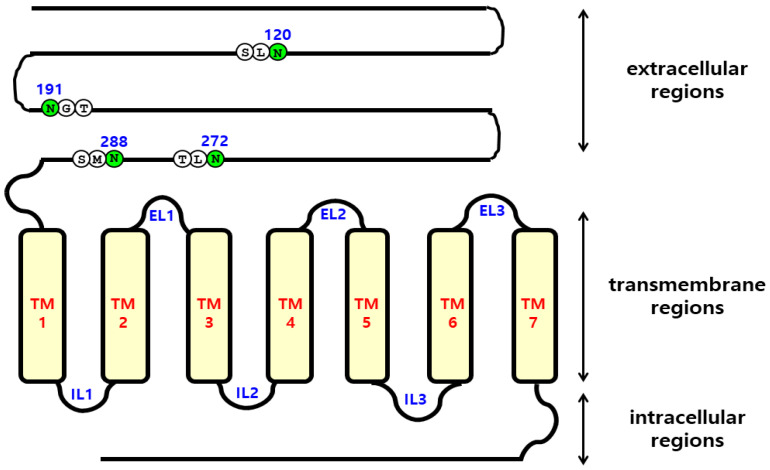
Schematic representation of the structure of eel FSHR. The N-linked glycosylation sites (N120, N191, N272, and N288) in the extracellular domain regions are indicated. The green circles indicate the putative glycosylated sites. TM, transmembrane domain; EL, extracellular loop; IL, intracellular loop.

**Figure 2 ijms-23-12792-f002:**
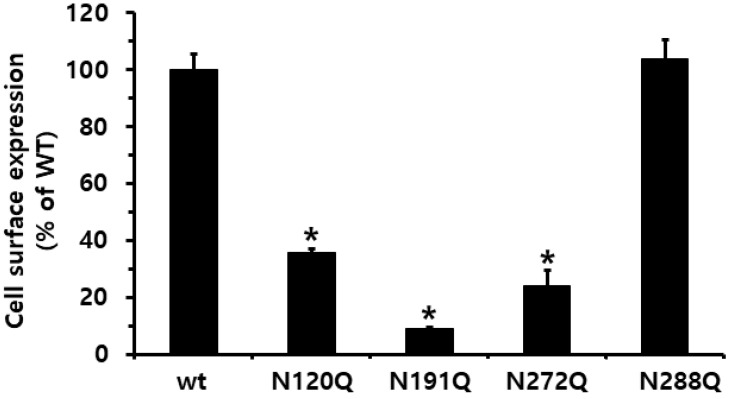
Cell surface expression of eel FSH receptors in transiently transfected HEK293 cells. Data are presented as the means ± SEM of three independent experiments. The surface expression in cells expressing eel FSHR-wt was considered at 100% (see Methods and Materials). * Statistically significant differences (*p* < 0.05) compared to the expression of the eel FSHR-wt.

**Figure 3 ijms-23-12792-f003:**
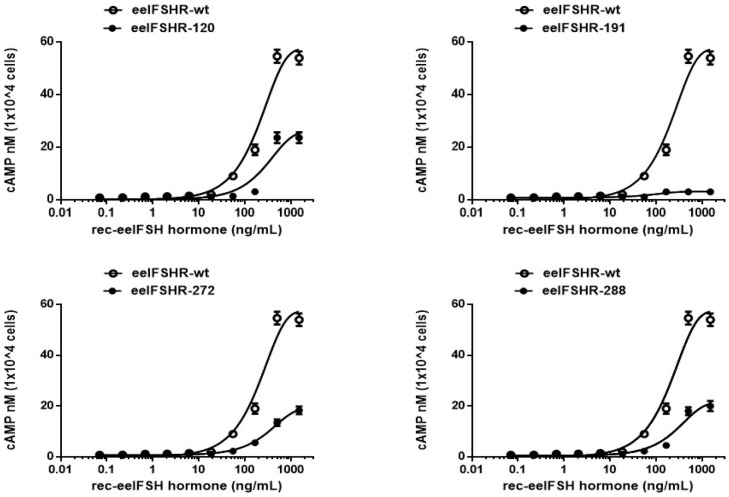
Total cAMP levels induced by stimulation with the recombinant eel FSH (rec- eel FSH) in CHO-K1 cells transiently transfected with the N-linked glycosylation site mutants of eel FSH. The empty circles denote eel FSHR-wt and the black circles denote mutants. The value of ΔF% was recalculated as cAMP concentration (nM). A representative dataset was obtained from three independent experiments. The figure depicts the results of the representative experiment performed with the indicated mutants.

**Figure 4 ijms-23-12792-f004:**
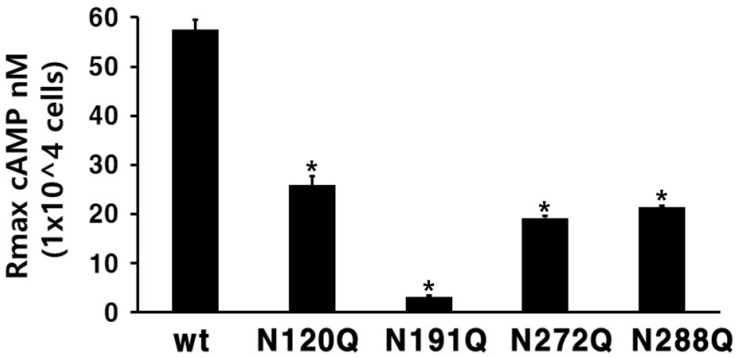
Rmax levels in the N-linked glycosylation mutants. The maximal cAMP responses presented in Figure 3 are displayed using a bar graph. * Statistically significant differences (*p* < 0.05) compared to the Rmax level of the eel FSHR-wt.

**Figure 5 ijms-23-12792-f005:**
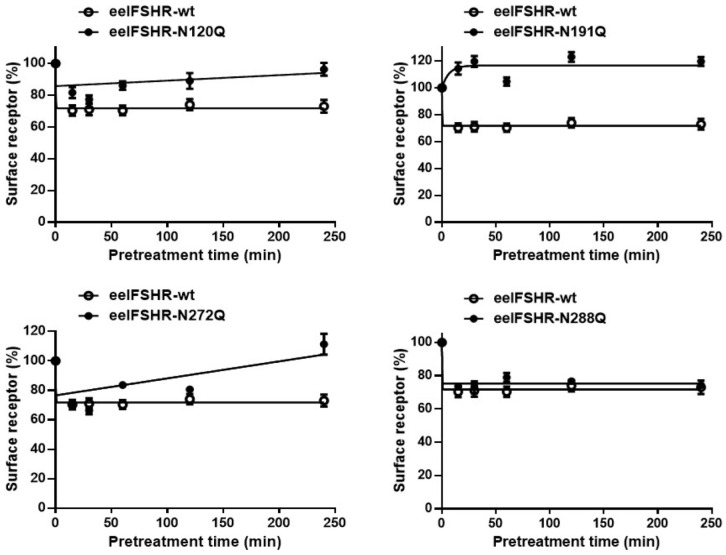
Time-dependent cell surface loss in the eel FSHR-wt and N-linked glycosylation mutants. Cell surface expression in the non-pretreated groups was taken as 100%. Mean data were fitted to the one-phase exponential decay equation (see Section 4). The blank circles are the same curves of eel FSHR-wt.

**Figure 6 ijms-23-12792-f006:**
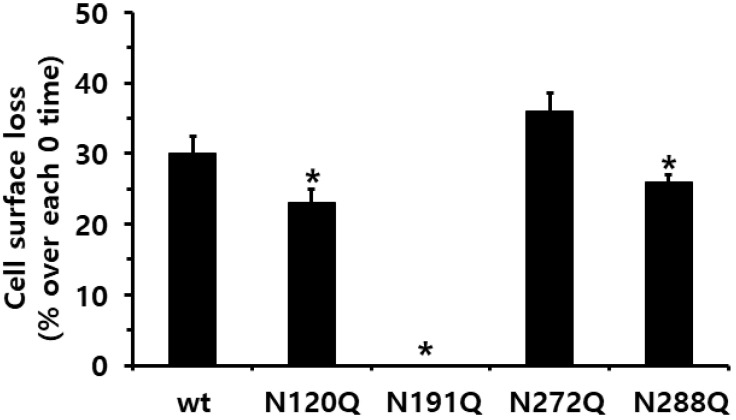
Cell surface loss of eel FSHR-wt and N-linked glycosylation mutants. The results of surface loss receptors are expressed as a percentage in the absence of agonist treatment (considered as 0% cell surface loss). * Statistically significant differences (*p* < 0.05) in the cell surface loss of the receptor compared to those of the eel FSHR-wt.

**Table 1 ijms-23-12792-t001:** Bioactivity of eel FSH receptors in cells expressing glycosylation site mutant receptors.

cAMP Responses			
eel FSH Receptors	Basal ^*a*^ (nM/10^4^ Cells)	EC_50_ (ng/mL)	Rmax ^*b*^ (nM/10^4^ Cells)
eel FSHR-wt	0.2 ± 0.1	201.1 (1.0-fold) (165.2 to 256.8) ^*c*^	57.5 ± 2.0 (100%)
eel FSHR- N120Q	0.1 ± 0.1	283.3 (0.70-fold) (199.2 to 490.4)	25.9 ± 1.9 (45.0%)
eel FSHR- N191Q	0.2 ± 0.1	- ^*d*^	3.1 ± 0.1 (0.05%)
eel FSHR- N272Q	0.3 ± 0.1	326.8 (0.61-fold) (284.2 to 384.6)	19.2 ± 0.4 (33.4%)
eel FSHR- N288Q	0.4 ± 0.2	281.0 (0.71-fold) (218.4 to 394.0)	21.4 ± 0.3 (37.2%)

Values are the means ± SEM of triplicate experiments. Log (EC_50_) values determined from the concentration-response curves from in vitro bioassays. ^*a*^ Basal cAMP level average without agonist treatment. ^*b*^ Rmax average cAMP level/10^4^ cells. ^*c*^ Geometric mean (95% confidence limit) of at least three experiments. ^*d*^ Not detected.

## Data Availability

Not applicable.

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
