# Peer review of "The N-Linked Glycosylation Site N191 Is Necessary for PKA Signal Transduction in Eel Follicle-Stimulating Hormone Receptor"

_ijms, 2022, doi:10.3390/ijms232112792_

Round 1

Reviewer 1 Report

This paper describes characterization of eel FSHR mutants that modified N-linked glycosylation sites. Since several changes in their properties have been reported, I judged that the paper could be published in the International Journal of Molecular Sciences, if the following points were appropriately revised.

1) In some parts of Introduction, Titles and Authors, spaces between characters are not inserted properly. Please correct them.

2On page 5, line 8 through 11, it says "We suggest that these glycosylation sites are involved in signal transduction, as indicated by the markedly decreased EC50 and Rmax levels of the mutants". But it is reasonable to assume that the reduction in cAMP production in the two mutants (N120Q and N272Q) is due to the low expression levels of the mutant proteins. Correction of description or addition of explanation is required.

3) In the followed part, it says "Specifically, the cAMP response of the N191Q mutant was completely inhibited, demonstrating that the signal transduction for cAMP responsiveness was not activated by high-concentration ligand treatment in cells expressing the N191Q mutant. Therefore, the N191 site in the N-linked glycosylation site of eel FSHR plays a pivotal role in cAMP response". However, since the N191Q mutant has a very low expression level, comparing its activity to that of the WT is unlikely to be worth discussing. Correction of description or addition of explanation is required.

4) Please make comprehensive revisions to the Discussion part based on the above comments, 2 and 3.

Author Response

Reviewer 1

Comments and Suggestions for Authors

This paper describes characterization of eel FSHR mutants that modified N-linked glycosylation sites. Since several changes in their properties have been reported, I judged that the paper could be published in the International Journal of Molecular Sciences, if the following points were appropriately revised.

1) In some parts of Introduction, Titles and Authors, spaces between characters are not inserted properly. Please correct them.

→We checked the “Titles and Authors” as reviewer comments.

2) On page 5, line 8 through 11, it says "We suggest that these glycosylation sites are involved in signal transduction, as indicated by the markedly decreased EC50 and Rmax levels of the mutants". But it is reasonable to assume that the reduction in cAMP production in the two mutants (N120Q and N272Q) is due to the low expression levels of the mutant proteins. Correction of description or addition of explanation is required.

→We inserted the “We insisted that two mutants (N120Q and N272Q) may be due to low expression level on the cell surface, but N288Q mutant was fully expressed on the cell surface.” in the Line 10-14, Page 5.

3) In the followed part, it says "Specifically, the cAMP response of the N191Q mutant was completely inhibited, demonstrating that the signal transduction for cAMP responsiveness was not activated by high-concentration ligand treatment in cells expressing the N191Q mutant. Therefore, the N191 site in the N-linked glycosylation site of eel FSHR plays a pivotal role in cAMP response". However, since the N191Q mutant has a very low expression level, comparing its activity to that of the WT is unlikely to be worth discussing. Correction of description or addition of explanation is required.

→We inserted the “The expression levels of the N191Q mutant were dramatically decreased to 9% compared to that of eel FSHR-wt. Thus, we also insist that the completely impaired cAMP response is involved in the expression level on cell surface” in the Line 16-19, Page 5.

4) Please make comprehensive revisions to the Discussion part based on the above comments, 2 and 3.

→We inserted the “We identified two mutants (N120Q and N272Q) for N-glycosylation, which suggest that modification is important for surface expression and functional response, indicating that those mutations were modified as inactive conformation. However, N228Q mutant resulted in dramatically decreased cAMP response despite fully expression on cell surface. Therefore, we suggest that the stability and conformation of the receptors are recognized as extremely important in the cAMP signal transduction. The specific site of Asn191 appears to be related to the signal transduction pathway. We also assume that the mutation at this site may result in a lower expression on the cell surface, suggesting that the cAMP response and cell-surface loss of the receptor are completely impaired.” in the line 33-41, Page 7.

Reviewer 2 Report

Byambaragchaa et al., through mutagenesis and activity assays have found four sites critical in PKA signal transduction in eel follicle-stimulating hormone receptor. The study is well -designed and intelligible and I recommend it for publication. However, authors need to check for any grammatical errors all through out the manuscript.

Author Response

Reviewer 2

Comments and Suggestions for Authors

Byambaragchaa et al., through mutagenesis and activity assays have found four sites critical in PKA signal transduction in eel follicle-stimulating hormone receptor. The study is well -designed and intelligible and I recommend it for publication. However, authors need to check for any grammatical errors all throughout the manuscript.

- We checked "grammatical contents of all sentences" as reviewer’ comments.

Reviewer 3 Report

Min and co workers describes study to determine whether conserved glycosylated sites in eel FSHR are necessary for cyclic adenosine monophosphate (cAMP) signal transduction. They used site-directed mutagenesis in the N-linked glycosylation sites of eel FSHR. In eel FSHR wild-type (wt), the cAMP response was gradually increased in a dose-dependent manner, displaying a high response at the Rmax level.

I recommend to accept the manuscript after some minor corrections in typos.

1. Page 2 line 74; replace 'we found' with 'we found' 

2. Page 2 line 76; 'l' in ligand should be in capital

3. Page 2 line 79; add proper spacing in ' These....dated'

4. Page 2 line 99; replace percentage presentation in 103% with times presentation. Percentage is used to present portion of a fraction and more than 100% of a finite capacity is not possible.

5. Page 2, line 103; ",,,,, expressing the mutant" please provide a reference in support of this statement.

6. Page 3, line 20; 57.5 nM/104 cells or 57.5 nM/104 cells; please check

7. Page 5 line 6; The sentence "In cells......... respectively' should be in same paragraph

Author Response

Reviewer 3

Comments and Suggestions for Authors

Min and co-workers described study to determine whether conserved glycosylated sites in eel FSHR are necessary for cyclic adenosine monophosphate (cAMP) signal transduction. They used site-directed mutagenesis in the N-linked glycosylation sites of eel FSHR. In eel FSHR wild-type (wt), the cAMP response was gradually increased in a dose-dependent manner, displaying a high response at the Rmax level.

I recommend to accept the manuscript after some minor corrections in typos.

  1. Page 2 line 74; replace 'we found' with 'we found' 

- We checked “the content” by reviewer’ comments. But this sentence might be by the PDF file modification.

  1. Page 2 line 76; 'l' in ligand should be in capital

- We changed “I” to “L” by reviewer’ comments

  1. Page 2 line 79; add proper spacing in ' These....dated'

- We checked “the content” by reviewer’ comments. But this sentence might be by the PDF file modification.

  1. Page 2 line 99; replace percentage presentation in 103% with times presentation. Percentage is used to present portion of a fraction and more than 100% of a finite capacity is not possible.

- The percentage (103%) is deleted in this line.

  1. Page 2, line 103; ",,,,, expressing the mutant" please provide a reference in support of this statement.

- We explained the results of N191Q mutant in this section. Thus, reference was no necessary.

  1. Page 3, line 20; 57.5 nM/104 cells or 57.5 nM/104cells; please check

- We changed "57.5 nM/104” to “57.5 nM/104 cells”

  1. Page 5 line 6; The sentence "In cells......... respectively' should be in same paragraph

-We changed “In cells….respectively” to “The EC50 values in cells expressing the N272Q and N288Q mutants were 326.8 ng/mL and 281 ng/mL, respectively.” in the Line 5-6 on Page 5.
